# Early Phase Increase in Serum TIMP-1 in Patients with Acute Encephalopathy with Biphasic Seizures and Late Reduced Diffusion

**DOI:** 10.3390/children10010078

**Published:** 2022-12-30

**Authors:** Ayaka Kasai, Jun Kobayashi, Makoto Nishioka, Noriko Kubota, Yuji Inaba, Mitsuo Motobayashi

**Affiliations:** 1Department of Medical Genetics, Shinshu University School of Medicine, 3-1-1 Asahi, Matsumoto, Nagano 390-8621, Japan; 2Clinical Laboratory, Nagano Children’s Hospital, 3100 Toyoshina, Azumino 399-8288, Japan; 3Life Science Research Center, Nagano Children’s Hospital, 3100 Toyoshina, Azumino 399-8288, Japan; 4Division of Neuropediatrics, Nagano Children’s Hospital, 3100 Toyoshina, Azumino 399-8288, Japan; 5Neuro-Care Center, Nagano Children’s Hospital, 3100 Toyoshina, Azumino 399-8288, Japan; 6Department of Pediatrics, Shinshu University School of Medicine, 3-1-1 Asahi, Matsumoto, Nagano 390-8621, Japan

**Keywords:** acute encephalopathy with biphasic seizures and late reduced diffusion, exanthem subitum, human herpesvirus-6, matrix metalloproteinase-9, tissue inhibitor matrix metalloproteinase 1

## Abstract

Background: Acute encephalopathy with biphasic seizures and late reduced diffusion (AESD) is the most frequent subtype of acute encephalopathy syndrome among Japanese children. Exanthem subitum is the most common causative infectious disease of AESD. We herein retrospectively analyzed serum and cerebrospinal fluid (CSF) concentrations of matrix metalloproteinase-9 (MMP-9), tissue inhibitor matrix metalloproteinase-1 (TIMP-1), and seven cytokines in patients with AESD or prolonged febrile seizure (FS) to assess the pathophysiology of AESD and detect biomarkers for diagnosing AESD in the early phase. Methods: Serum and CSF samples were obtained from 17 patients with AESD (1st seizure phase group, *n* = 7; 2nd seizure phase group, *n* = 10) and 8 with FS. The concentrations of MMP-9, TIMP-1, and seven cytokines were measured by enzyme-linked immunosorbent assays or cytometric bead arrays. Results: Serum concentrations of TIMP-1 were significantly higher in the 1st seizure phase group than in the 2nd seizure phase group. No significant differences were observed in serum concentrations of MMP-9 or the MMP-9/TIMP-1 ratio. Conclusions: The MMP-9-independent increase in circulating TIMP-1 concentrations observed in the present study may be associated with the pathophysiology of AESD in the 1st seizure phase.

## 1. Introduction

Acute encephalopathy with biphasic seizures and late reduced diffusion (AESD) is a type of acute encephalopathy characterized by biphasic convulsions and impaired consciousness that is induced by infection [1]. AESD is the most common subtype of acute encephalopathy syndrome in Japanese children [1]. The median age of onset is 1 year and the annual number of cases is 100–200 [1]. Although the pathophysiology of AESD may be excitotoxic disorders associated with late-onset neuronal cell death, this has yet to be confirmed [2,3]. Human herpesvirus-6 (HHV-6) is the most common causative agent of AESD, followed by the influenza virus and HHV-7 [1]. There are currently no established biomarkers for diagnosing AESD or predicting the severity of neuronal damage in the early acute phase, which has been described as “the 1st seizure phase”.

Matrix metalloproteinase-9 (MMP-9), a member of the endopeptidase family [4], has been reported to disrupt the blood–brain barrier (BBB) by degrading all components of the extracellular matrix, including laminin, collagen, and fibronectin [5]. Tissue inhibitor matrix metalloproteinase-1 (TIMP-1) is a glycoprotein that regulates the degradation of the extracellular matrix by inhibiting the enzymatic activity of MMP-9 [6]. An increase in circulating MMP-9 degrades the BBB, whereas an increase in serum TIMP-1 prevents this degradation. Therefore, an increased MMP-9/TIMP-1 ratio has been suggested to impair the BBB, which ultimately leads to encephalopathy [7]. On the other hand, an increase in serum concentrations of TIMP-1, but not MMP-9 of the MMP-9/TIMP-1 ratio has been reported in patients with central nervous disorders, such as cerebral infarction, cranial hemorrhage, traumatic brain injury, hemolytic-uremic syndrome, and HHV-6 encephalitis [2,8,9,10,11,12]. Therefore, the role of TIMP-1 needs to be considered based on these two aspects, i.e., the MMP-9/TIMP-1 balance and MMP-9-independent increases in TIMP-1. However, there is currently no evidence to show elevated concentrations of TIMP-1 in patients with AESD.

In the present study, we retrospectively investigated serum and cerebrospinal fluid (CSF) concentrations of MMP-9 and TIMP-1 in patients with AESD or prolonged febrile seizure (FS). We also analyzed seven types of cytokines, i.e., interleukin (IL)-1β, -2, -4, -10, -17A, and -21 and macrophage inflammatory protein-1 alpha (MIP-1α), which may be involved in TIMP-1-induced inflammation. CD4+ T cell- and monocyte/macrophage-related cytokines were targeted in the present study based on previous findings [13,14,15,16,17,18,19,20,21]. This is the first study to report MMP-9-independent increases in serum concentrations of TIMP-1 in patients with AESD in the 1st seizure phase over those in the 2nd seizure phase, which may be partly responsible for the pathophysiology of AESD.

## 2. Materials and Methods

### 2.1. Subjects and Sample Collection

Seventeen patients with AESD (11 males and 6 females) and 8 with prolonged FS (4 males and 4 females) treated at Nagano Children’s Hospital between 4 April 2014 and 9 July 2020 were retrospectively enrolled. Patients were diagnosed with AESD by pediatric neurologists based on standard clinical criteria: (1) convulsions, mostly status epilepticus, 24 h or less after the onset of fever; (2) the transient amelioration of impaired awareness; (3) the reappearance of convulsions, mostly clustered focal seizures, followed by disturbed consciousness on the fourth to sixth day after onset; (4) causative pathogens of antecedent infection, mainly influenza virus or HHV-6/7; (5) a wide range of neurological prognoses; (6) normal magnetic resonance imaging (MRI) results within two days of the onset; and (7) the bright tree appearance on cerebral MRI: high-intensity lesions in the subcortical white matter, called U-fibers, on diffusion-weighted images on the third to the ninth day of illness [1]. Serum and CSF samples (serum, *n* = 21; CSF, *n* = 23) were collected in pairs from 19 patients on admission. A CSF sample was only obtained from 4 patients and a serum sample solely from 2. All 25 patients were divided into three groups. Patients in “the 1st seizure phase group” (serum, *n* =7; CSF, *n* = 6) and “the 2nd seizure phase group” (serum *n* = 8, CSF *n* = 9) presented to our hospital in the 1st and 2nd seizure phases of AESD, respectively. “The prolonged FS group” (serum, *n* = 6; CSF, *n* = 8) included patients who developed prolonged FS. FS was defined as seizures provoked by fever ≥38.0 °C according to a previous study, and a prolonged seizure was defined as a seizure lasting more than 30 min or recurrent seizures persisting for a total of more than 30 min without fully recovering consciousness [22]. Among patients in the prolonged FS group, those with other neurological disorders, such as encephalitis, encephalopathy, meningitis, brain abscess, brain tumor, and epilepsy, were excluded based on the clinical course, a blood examination, CSF test, electroencephalogram, and brain MRI. All patients in the prolonged FS group regained consciousness within 24 h without neurological sequelae. All samples were collected on admission and stored at −80 °C for later analyses.

### 2.2. MMP-9 and TIMP-1 Assays

The serum and CSF concentrations of MMP-9 and TIMP-1 were measured using enzyme-linked immunosorbent assay (ELISA) kits (R&D Systems, Minneapolis, MN, USA). The detectable range for each ELISA kit was 0.313 to 20 ng/mL and 0.156 to 10 ng/mL, respectively. The ELISA kit for MMP-9 used in the present study measures the 92 kDa pro- and 82 kDa active forms of MMP-9. Measurements were performed according to the manufacturer’s instructions. Samples were assayed in triplicate and average values were adopted. All assays were completed in one day.

### 2.3. Cytokine Assay

Serum and CSF concentrations of IL-1β, -2, -4, -10, -17A, and -21 and MIP-1α were measured using the BD™ Cytometric Bead Array^®^ (Becton, Dickinson, and Company [BD], East Rutherford, NJ, USA) and BD FACSCanto™ II Flow Cytometer (BD) equipped with FCAP Array™ v3.0 Software (BD, East Rutherford, NJ, USA) according to the manufacturer’s instructions. The detectable ranges for IL-1β, -2, -4, -10, and -17A and MIP-1α were 10 to 2500 pg/mL, while that for IL-21 was 40 to 10,000 pg/mL. Measurements were performed according to the manufacturer’s instructions. Samples were analyzed once. All assays were completed in one day.

### 2.4. Statistical Analysis

Values are presented as the median (range). Statistical comparisons of sex and the presence of a prolonged seizure of more than 40 min were performed using the chi-square test, and other clinical and laboratory findings and cytokine profiles were assessed by the Steel–Dwass method. To evaluate the significance of differences in TIMP-1 and MMP-9 concentrations among the three independent groups, the Tukey–Kramer method was used when data were parametric and the Steel–Dwass method when data were non-parametric. The relationship between the two parameters was analyzed by Spearman’s correlation test. All statistical analyses were conducted using Microsoft^®^ Excel^®^ for windows version 2016, Statcel 4 software (OMS Publishing Inc., Saitama, Japan). Cytokine levels below the lower detectable range were defined as 1.0 pg/mL for statistical analyses. The level of significance was defined as a *p* value < 0.05.

### 2.5. Ethics

The present study was approved by the Ethics Committee of the Nagano Children’s Hospital (approval number: 31-11; approval date: 9 September 2019).

## 3. Results

### 3.1. Clinical Characteristics

The clinical data and laboratory findings of the 1st and 2nd seizure phase groups and prolonged FS group are shown in Table 1. No significant differences were observed in age, sex, or the number of patients with a prolonged seizure of more than 40 min. Laboratory data, such as blood values of aspartate aminotransferase, creatinine, and sugar and CSF levels of the cell count, total protein, and lactate, did not significantly differ among the three groups. On the other hand, a significant difference was observed in neuron-specific enolase (NSE) in the CSF and C-reactive protein (CRP). The CSF concentrations of NSE in the 1st and 2nd seizure phase groups and prolonged FS group were 12.6 (range, 6.0–61.0), 81.1 (range, 17.1–366.0), and 7.8 (range, 6.3–13.2) ng/mL, respectively. Significant differences were observed between the 1st and 2nd seizure phase groups (*p* < 0.05) and between the 2nd seizure phase group and prolonged FS group (*p* < 0.01) (Table 1 and Appendix A). In addition, CRP levels were significantly higher in the 1st seizure phase group (0.90 mg/dL; range, 0.38–5.11) than in the 2nd seizure phase (0.30 mg/dL; range, 0.08–1.66). The localization of the bright tree appearance on brain MRI was as follows: three patients each from the 1st and 2nd seizure phase groups had a bilateral widespread lesion, one from the 1st seizure phase group and four from the 2nd seizure phase group had bilateral frontal lobe lesions, three from the 1st seizure phase group and one from the 2nd seizure phase group had a unilaterally widespread lesion, and two from 2nd seizure phase group with a unilaterally frontal lobe lesion. All patients in the 1st and 2nd seizure phase groups received 72 h of targeted temperature management and were administered of dextromethorphan and vitamins. Six patients from the 1st seizure phase group and all patients from the 2nd seizure phase group were administered edaravone. Methylprednisolone pulse therapy was initiated for four patients from the 1st seizure phase group and one from the 2nd seizure phase group, and intravenous immunoglobulin was administered to one each from the 1st and 2nd seizure groups. Five patients each in the 1st and 2nd seizure phage groups had residual neurologic sequelae, including intellectual disability, developmental disability, and epilepsy. Ten (58.8%) out of the seventeen patients with AESD in the 1st and 2nd seizure groups had neurological sequelae.

### 3.2. Serum Concentrations of MMP-9 and TIMP-1

Serum TIMP-1 concentrations were significantly higher in the 1st seizure phase group (289.0 ng/mL; range, 146.6–390.9) than in the 2nd seizure phase (140.9 ng/mL; range, 108.3–212.7) (*p* < 0.05) (Figure 1B). Significant differences were not observed in serum MMP-9 concentrations or MMP-9/TIMP-1 ratios among the three groups (Figure 1A,C).

### 3.3. CSF Concentrations of MMP-9 and TIMP-1

As shown in Figure 2A, CSF concentrations of MMP-9 were below the lower detection limit, while TIMP-1 concentrations and MMP-9/TIMP-1 ratios in CSF were not significantly different among the three groups (Figure 2B,C).

### 3.4. Relationship between Serum Concentrations of MMP-9 and TIMP-1

A positive correlation was observed between serum concentrations of MMP-9 and TIMP-1 (*p* < 0.05, Rs = 0.474) (Figure 3A), whereas no interaction was detected by a group-based analysis (Figure 3B–D).

### 3.5. Relationship between Serum TIMP-1 and CSF NSE Concentrations

Serum concentrations of TIMP-1 did not correlate with CSF concentrations of NSE (Appendix A).

### 3.6. Relationship between Serum Concentrations of TIMP-1 and Serum or CSF Concentrations of Seven Types of Cytokines

Serum concentrations of TIMP-1 did not correlate with serum or CSF concentrations of seven types of cytokines (IL-1β, -2, -4, -10, -17A, and -21 and MIP-1α).

### 3.7. Cytokine Profiles in Serum and CSF

As shown in Table 2, serum and CSF concentrations of seven types of cytokines (IL-1β, -2, -4, -10, -17A, and -21 and MIP-1α) did not significantly differ in patients with AESD or prolonged FS among the three groups.

## 4. Discussion

We retrospectively analyzed clinical characteristics and serum and CSF concentrations of MMP-9, TIMP-1, and seven types of cytokines in children with AESD. Serum concentrations of TIMP-1 were significantly higher in the 1st seizure phase group than in the 2nd seizure phase group. Serum concentrations of MMP-9 and MMP-9/TIMP-1 ratios did not significantly differ among the three groups. CSF concentrations of NSE were significantly higher in the 2nd seizure phase group than in the prolonged FS group, which may reflect subacute neuronal damage caused by AESD [23]. CSF concentrations of NSE did not correlate with serum concentrations of TIMP-1.

The role of TIMP-1, the main factor examined in the present study, needs to be discussed based on two aspects, i.e., the MMP-9/TIMP-1 balance and MMP-9-independent increases in TIMP-1. MMPs are a family of zinc- and calcium-dependent endopeptidases composed of 24 known subtypes, ranging from MMP-1 to -28 [24]. MMPs are classified into the following five categories according to the domain composition and substrate preference: collagenases, gelatinases, stromelysins, matrilysins, and membrane types. MMP-9 is a gelatinase that digests components of the extracellular matrix, including collagen, laminin, and fibronectin [24]. Therefore, increases in serum concentrations of MMP-9 may be destructive to the BBB because the basement membrane of the BBB is composed of these matrixes [7]. On the other hand, TIMPs are a family of endogenous inhibitors of MMPs comprised of four subgroups: TIMP-1 to -4 [6,25]. The enzymatic activity of MMP-9 is inhibited by one-to-one non-covalent binding with TIMP-1; therefore, TIMP-1 may protect the BBB from destruction by MMP-9. Previous studies reported elevated concentrations of MMP-9 and/or a high MMP-9/TIMP-1 ratio in the serum of patients with central nervous diseases, such as intracerebral hemorrhage, traumatic brain injury, multiple sclerosis, and febrile convulsive disorders, including FS and acute encephalitis or encephalopathy [7,9,11,26,27]. Ichiyama et al. demonstrated that serum concentrations of MMP-9 and the MMP-9/TIMP-1 ratio increased in influenza-associated encephalopathy with a poor prognosis [7]. In children with AESD, Suenaga et al. showed that serum concentrations of MMP-9 were significantly higher in patients with AESD than in those with simple FS or epilepsy and controls; serum concentrations of TIMP-1 in AESD were significantly lower than those in patients with FS or epilepsy and controls [22]. However, these findings need to be carefully interpreted because this study did not describe the clinical phase of AESD, i.e., the 1st or 2nd seizure phase.

In contrast to these findings, no significant increases were observed in serum concentrations of MMP-9 or the MMP-9/TIMP-1 ratio in the present study, suggesting that the increase in circulating TIMP-1 was independent of MMP-9. Leonard et al. reported an MMP-9-independent increase in TIMP-1 concentrations in patients with cerebral infarction [10], and Shiraishi et al. detected elevated TIMP-1 concentrations in patients with encephalopathy due to hemolytic uremic syndrome [8]. However, the source or cause of the MMP-9-independent increase in TIMP-1 in these studies and the present study remains unclear. We have two hypotheses: the innate immune response following viral infection and TIMP-1-secreting cells. Innate immunity is mainly controlled by dendritic cells, natural killer cells [20], and monocytes/macrophages [15,19,21], and these cells have been proposed as candidate cells for the secretion of TIMP-1 in vitro [13]. In vivo, patients with sepsis were found to have high TIMP-1 and low MMP-9 concentrations in serum, which predicted a monocyte-derived increase in TIMP-1 concentrations in peripheral blood [16,17]. Therefore, these immune cells or related cytokines may be involved in the MMP-9-independent increase in TIMP-1 and the development of AESD. In addition, HHV-6 and -7, which are causative viruses of exanthem subitum, infect several immunocytes, including monocytes/macrophages and CD4+ T cells, the latter of which have been identified as a candidate secreting TIMP-1 in vitro [15,18,19,21]. In children with AESD due to exanthem subitum, Kawamura et al. analyzed cytokine and chemokine profiles in serum and CSF: significant increases in serum concentrations of IL-10 and a significant decrease in CSF concentrations of IL-1β were observed [28]. Additionally, a few studies or case reports showed the increases of serum concentrations of IL-6 and -10 and tumor necrosis factor-α in patients with AESD, although the cause of increases of these cytokines had not been demonstrated [27,29,30,31]. Based on these findings, we herein focused on seven types of cytokines related to CD4+ T cells and monocytes/macrophages (IL-1β, -2, -4, -10, -17A, and -21 and MIP-1α). However, no significant differences were observed in any of the seven cytokines among the three groups, and, thus, further analyses are warranted with a larger study design.

Furthermore, an imbalance in the MMP-9/TIMP-1 system may have affected the development of AESD in children in the present study. A positive correlation was observed between serum concentrations of MMP-9 and TIMP-1 (Figure 3A), although there was no significant increase in circulating concentrations of MMP-9 or the MMP-9/TIMP-1 ratio (Figure 1). Therefore, damage to the BBB based on an imbalance in MMP-9/TIMP-1 may partially contribute to the development of AESD, which is consistent with previous findings [22].

## 5. Conclusions

An MMP-9-independent increase in serum concentrations of TIMP-1 may be associated with the pathophysiology of AESD in the 1st seizure phase. The source or cause of the increase in serum concentrations of TIMP-1 remains unclear and, thus, further studies with a large number of patients are warranted.

## Figures and Tables

**Figure 1 children-10-00078-f001:**
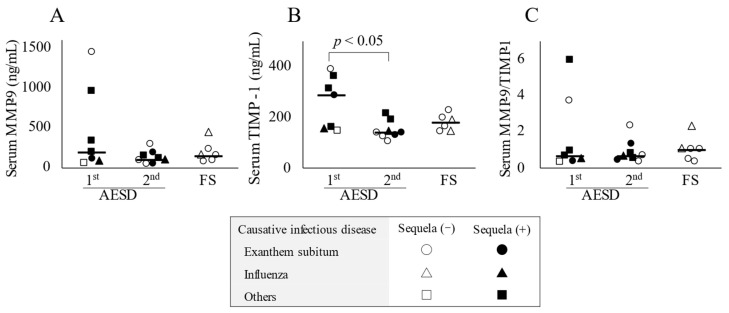
Serum concentrations of biomarkers in patients with acute encephalopathy with biphasic seizures and late reduced diffusion (AESD) in the 1st or 2nd seizure phase or with prolonged febrile seizure (FS). Serum concentrations of matrix metalloproteinase-9 (MMP-9) (**A**) and tissue inhibitor matrix metalloproteinase-1 (TIMP-1) (**B**) and the MMP-9/TIMP-1 ratio in serum (**C**) in all patients with AESD or prolonged FS. The horizontal bar represents the median value. Statistical comparisons by the Tukey–Kramer or Steel–Dwass test. Circles, triangles, and squares represent patients with AESD or FS due to exanthem subitum, influenza, and other infectious diseases, respectively. Filled marks donate patients with sequela (e), and open marks without sequela.

**Figure 2 children-10-00078-f002:**
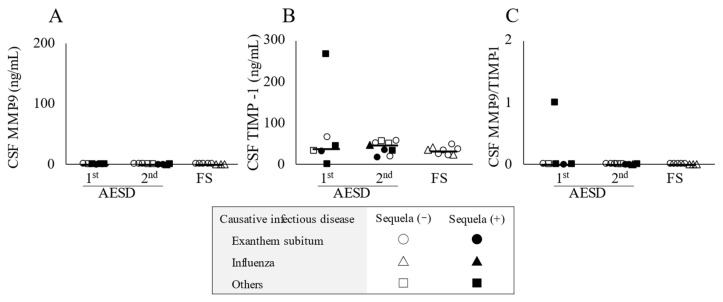
Cerebrospinal fluid (CSF) concentrations of the biomarkers in patients with acute encephalopathy with biphasic seizures and late reduced diffusion (AESD) in the 1st or 2nd seizure phase or with prolonged febrile seizure (FS). CSF levels of matrix metalloproteinase-9 (MMP-9) (**A**) and tissue inhibitor matrix metalloproteinase-1 (TIMP-1) (**B**) and the MMP-9/TIMP-1 ratio in CSF (**C**) in all patients with AESD or prolonged FS. Statistical comparisons by the Tukey–Kramer or Steel–Dwass test. Circles, triangles, and squares represent patients with AESD or FS due to exanthem subitum, influenza, and other infectious diseases, respectively. Filled marks donate patients with sequela (e), and open marks without sequela.

**Figure 3 children-10-00078-f003:**
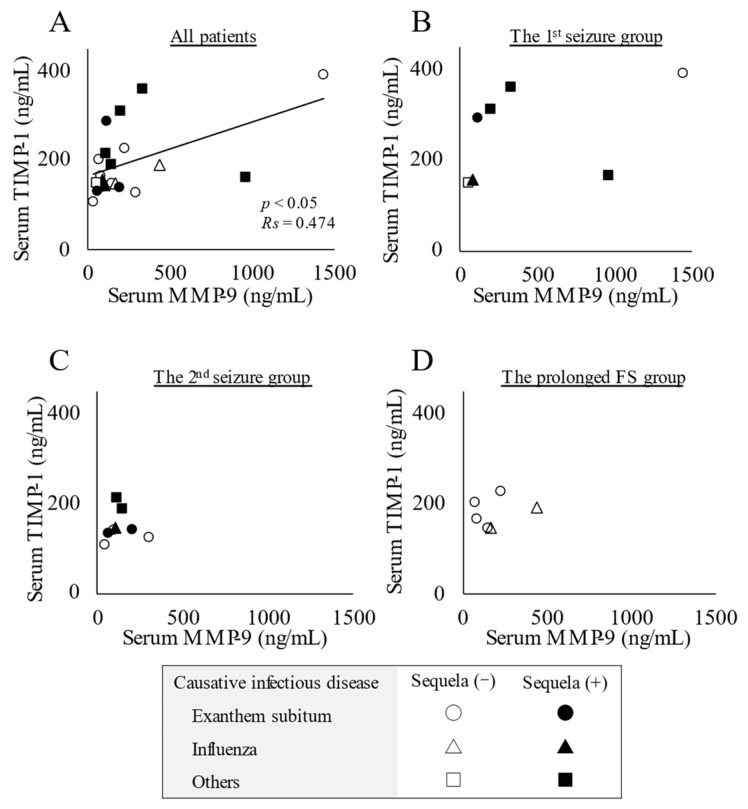
Relationship between serum concentrations of matrix metalloproteinase-9 (MMP-9) and tissue inhibitor matrix metalloproteinase-1 (TIMP-1). (**A**) Positive correlation between serum concentrations of MMP-9 and TIMP-1 in all patients with acute encephalopathy with biphasic seizures and late reduced diffusion (AESD) or prolonged febrile seizure (FS). Statistical comparisons by Spearman’s correlation test. (**B**) The 1st seizure phase group. (C) The 2nd seizure phase group. (**D**) The prolonged FS group. Circles, triangles, and squares represent patients with AESD or FS due to exanthem subitum, influenza, and other infectious diseases, respectively. Filled marks donate patients with sequela (e), and open marks without sequela.

**Table 1 children-10-00078-t001:** Clinical and laboratory findings of patients with acute encephalopathy with biphasic seizures and late reduced diffusion (AESD) or prolonged febrile seizure (FS).

	AESD		Prolonged FS Group Median (Range) or Number
	1st Seizure Phase Group Median (Range) or Number	2nd Seizure Phase Group Median (Range) or Number	
*n*	7	10	8
Age, years	1.8 (0.8–4.5)	1.8 (1.0–4.3)	1.8 (1.1–4.4)
Male/Female	5/2	6/4	4/4
Causative infectious disease			
Exanthema subitum	2	5	5
Influenza	1	1	3
Others	4	4	0
Underlying disorder			
Febrile seizure	1	2	2
Tuberous sclerosis	1	0	0
Corpus callosum hypoplasia	1	0	0
Migratory testis	0	1	0
Prader-Willi syndrome	0	0	1
Congenital hypothyroidism	0	0	1
None	4	7	4
Number of patients with a prolonged seizure of more than 40 min	4	3	1
Blood examinations			
Aspartate aminotransferase, U/L	51 (38–299)	48 (38–382)	46 (41–63)
Creatinine, mg/dL	0.34 (0.21–0.45)	0.26 (0.13–0.39)	0.30 (0.17–0.39)
Blood sugar, mg/dL	110 (82–164)	116 (85–196)	107 (93–297)
C-reactive protein, mg/dL	0.90 (0.38–5.11) *	0.30 (0.08–1.66)	0.31 (0.04–2.17)
Cerebrospinal fluid examinations			
Cell counts, /µL	1.4 (0.7–490.7)	1.3 (0.0–3.0)	0.5 (0.0–3.0)
Total protein, mg/dL	16 (7–114)	17 (6–172)	13 (8–22)
Neuron specific enolase, ng/mL	12.6 (6.0–61.0)	81.1 (17.1–366.0) ^†^	7.8 (6.3–13.2)
Lactate, mg/dL	17.8 (12.3–35.8)	10.6 (8.9–17.5)	13.3 (11.7–15.7)
Localization of the bright tree appearance on brain MRI			
Bilaterally widespread	3	3	0
Bilateral frontal lobes	1	4	0
Unilaterally widespread	3	1	0
Unilaterally frontal lobe	0	2	0
None	0	0	8
Treatment			
TTM	7	10	1
Dextromethorphan	7	10	4
Vitamin B1/B6/levocarnitine	7	10	3
Edaravone	6	10	1
Methylprednisolone pulse	4	1	0
Intravenous immunoglobulin	1	1	0
Acyclovir	0	0	1
None	0	0	4
Sequela			
ID/DD	2	4	0
Epilepsy	0	1	0
ID/DD and Epilepsy	3	0	0
None	2	5	8

ID/DD intellectual disability and/or developmental disability, MRI: magnetic resonance imaging, *n*: number of patients, TTM: targeted temperature management. * Significant difference between the 1st and 2nd seizure phase groups (*p* < 0.05). ^†^ Significant difference between the 2nd seizure phase group and the 1st seizure phase group (*p* < 0.05) and complex FS group (*p* < 0.01).

**Table 2 children-10-00078-t002:** Serum and cerebrospinal fluid (CSF) cytokine profiles in patients with acute encephalopathy with biphasic seizures and late reduced diffusion (AESD) or prolonged febrile seizure (FS).

	AESD		Prolonged FS Group Median (Range) or Number
	1st Seizure Phase Group Median (Range) or Number	2nd Seizure Phase Group Median (Range) or Number	
*n*	7	10	8
Serum			
IL-1β, pg/mL	N.D. (N.D.–18.8)	N.D.	N.D.
IL-2, pg/mL	N.D.	N.D.	N.D.
IL-4, pg/mL	N.D.	N.D.	N.D.
IL-10, pg/mL	28.7 (N.D.–125.4)	31.2 (N.D.–74.1)	41.7 (N.D.–72.0)
IL-17A, pg/mL	N.D. (N.D.–28.6)	N.D.	N.D.
IL-21, pg/mL	N.D. (N.D.–588.4)	N.D.	N.D. (N.D.–230.9)
MIP-1α, pg/mL	N.D. (N.D.–28.5)	N.D.	N.D.
CSF			
IL-1β, pg/mL	N.D.	N.D.	N.D.
IL-2, pg/mL	N.D.	N.D. (N.D.–79.2)	N.D.
IL-4, pg/mL	N.D.	N.D. (N.D.–33.1)	N.D.
IL-10, pg/mL	N.D. (N.D.–948.6)	N.D.	N.D.
IL-17A, pg/mL	N.D.	N.D. (N.D.–12.6)	N.D. (N.D.–11.8)
IL-21, pg/mL	N.D.	N.D. (N.D.–173.3)	N.D. (N.D.–194.3)
MIP-1α, pg/mL	N.D.	N.D.	N.D.

IL: interleukin, MIP-1α: macrophage inflammatory protein -1 alpha, *n*: number of patients, N.D.: all samples were below the detectable limit.

## Data Availability

Not applicable.

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
