# Peer review of "Early Phase Increase in Serum TIMP-1 in Patients with Acute Encephalopathy with Biphasic Seizures and Late Reduced Diffusion"

_children, 2022, doi:10.3390/children10010078_

Round 1

Reviewer 1 Report

The study by Kasai et al., investigates the relationships between MMP9 and TIMP1 with cytokines in patients with AESD and FS.  The authors’ data demonstrate an increase of circulating TIMP1 but not MMP9 in AESD compared to FS patients.  Cytokine values were likely not a significant contributor to this finding.  Overall, the strength of this data are the interesting results in particular regarding that the TIMP increase is MMP9-independent in AESD.  The weaknesses are the small sample size and many quantitative analyses coming back undetected, particularly for the cytokine analyses.  The authors need to address the following points:

1)    The sample size of this study is very small especially for the authors to separate into the exanthem subitem subgroups.

2)    I have some serious concerns regarding the cytokine analyses.  Many values are not detected and then values are set to 1.0 pg/ml.  This would not be an issue except it appears that more of the values are undetected than detected based on Table 2 and Figure 4.  This makes any analysis or conclusions based on this data not viable in this reviewer’s opinion.  If serum/CSF is banked, the authors should repeat the assay with more sensitive ELISA kits or use other methodology as necessary.

3)    Based on the previous statement, lines 343-346 is an overstatement with the acknowledged small sample size and lack of significance.

4)    NSE abbreviation is not defined.

5)    The y-axis for figure 2A, 2B, and 2C should be lowered so that individual values can be better distributed.  For example, figure 2A should range from 0-250, 2A should range from 0-400, and 2C should range from 0-2.

Author Response

Responses to the comments of Reviewer 1

We appreciate your critical and constructive comments, which have helped us to improve the quality of the revised manuscript.

  • The sample size of this study is very small especially for the authors to separate into the exanthem subitem subgroups.

Response: Thank you for raising this point. We agree that the sample size of our study was small, particularly in the subgroup analysis of patients with exanthem subitum. The main reason for the small sample size is the rarity of collecting blood and cerebrospinal fluid (CSF) samples from patients with AESD in the very acute phase. Nevertheless, we consider the results obtained to provide novel insights into this field because this is the first study to demonstrate an increase in the serum concentration of TIMP-1 only.

Furthermore, regarding the Reviewer’s comment, a subcategory analysis of patients with exanthem subitum is considered to be insufficient for reaching concrete conclusions. Therefore, we deleted the subcategory analysis from the revised manuscript. Specifically, we deleted the following sentences and/or Figures from the manuscript: “In additionally, subcategory analysis … in the 2nd seizure phase or FS groups” in the Abstract, “… and may be a diagnostic marker for distinguishing AESD from FS caused by exanthem subitem” in the Abstract, “Furthermore, subcategory analysis based on … in the early acute phase” in the Introduction, “Subcategory analysis in patients with … among the three groups (Figure 1D and 1F)” in the Results and Figure 1D and 1F, “Subcategory analysis in patients with … in the 2nd seizure phase group (all samples were below the detectable limit) (P < 0.05) (Table 2 and Figure 4D–4F)” in the Results and Figure 4D–4F, “Subcategory analysis based on … cytokines among the three groups” in the Discussion, “On the other hand, subcategory analysis … include children with exanthem subitem” in the Discussion, and “In addition, subcategory analysis … in patients with exanthem subitem” in the Conclusions.

  • I have some serious concerns regarding the cytokine analyses. Many values are not detected and then values are set to 1.0 pg/ml. This would not be an issue except it appears that more of the values are undetected than detected based on Table 2 and Figure 4. This makes any analysis or conclusions based on this data not viable in this reviewer’s opinion. If serum/CSF is banked, the authors should repeat the assay with more sensitive ELISA kits or use other methodology as necessary.

Response: Thank you for raising these important points. As indicated, we ideally need to repeat the assay with more sensitive ELISA kits or use another methodology if serum/CSF is banked. However, we were unable to attempt these approaches for a number of reasons, the main issue being the stocked sample volumes of patients. We depleted many of the samples collected from the enrolled patients, and, thus, are unable to repeat assays. Nevertheless, we consider the results obtained to provide novel insights into this field that will prompt future studies using more precise and sensitive methodologies to analyze cytokines.

  • Based on the previous statement, lines 343-346 is an overstatement with the acknowledged small sample size and lack of significance.

Response: We agree with this comment as mentioned in 1). We deleted lines 343-346 from the manuscript.

  • NSE abbreviation is not defined.

Response: We spelled out NSE in the text (line 165).

  • The y-axis for figure 2A, 2B, and 2C should be lowered so that individual values can be better distributed. For example, figure 2A should range from 0-250, 2A should range from 0-400, and 2C should range from 0-2.

Response: Thank you for your advice. The reason for the range of the y-axis in Figures 2A, 2B, and 2C being set at the submitted levels is for alignment with Figure 1. However, we agree with your advice and revised the ranges of the y-axis in Figure 2 as follows:

From: 0 to 1500 ng/mL in Figure 2A, 0 to 600 ng/mL in Figure 2B, and 0 to 6 in Figure 2C.

To: 0 to 200 ng/mL in Figure 2A, 0 to 300 ng/mL in Figure 2B, and 0 to 2 in Figure 2C.

Reviewer 2 Report

In this paper, the authors performed a retrospective analysis on both serum and cerebrospinal fluid levels of MMP-9, TIMP-1 and even cytokines in patients affected by acute encephalopathy with biphasic seizures and late reduced diffusion (AESD). I think that the manuscript can provide interesting and informative results in clinical neurology as potential biomarkers. However, this paper has the limit of sample size, although the authors are aware of it. Here, I report the major points to be addressed:

Major concerns include:

- Introduction: the authors should explain why they chose to analyze only these 7 cytokines and report the state of art in AESD.

- Materials and methods: did you measure the 92 kDa Pro- and 82 kDa active forms of MMP9?

- Results: I suggest the authors to better organize the tables/figures reporting only the significant findings.

- Discussion: the authors should better discuss their results in a more organized manner.

Minor concerns include:

- Please correct typos and English language throughout the manuscript (mainly in the discussion).

Author Response

Responses to the comments of Reviewer 2

We appreciate your critical and constructive comments, which have helped us to improve the quality of the revised manuscript.

  • Introduction: the authors should explain why they chose to analyze only these 7 cytokines and report the state of art in AESD.

Response: We appreciate this advice, according to which we revised the following description and references (lines 62-67):

From: “In the present study, we retrospectively investigated both serum and cerebrospinal fluid (CSF) levels of MMP-9, TIMP-1, and seven types of cytokines that may be involved in TIMP-1-related inflammation in patients with AESD or prolonged febrile seizure (FS).”

To: “In the present study, we retrospectively investigated serum and cerebrospinal fluid (CSF) concentrations of MMP-9 and TIMP-1 in patients with AESD or prolonged febrile seizure (FS). We also analyzed seven types of cytokines, i.e., interleukin (IL)-1β, -2, -4, -10, -17A, and -21 and macrophage inflammatory protein-1 alpha (MIP-1α), which may be involved in TIMP-1-induced inflammation. CD4+ T cell- and monocyte/macrophage-related cytokines were targeted in the present study based on previous findings [13-21].”

  • Materials and methods: did you measure the 92 kDa Pro- and 82 kDa active forms of MMP9?

Response: The ELISA kit for MMP-9 used in the present study measures the 92-kDa pro- and 82-kDa active forms of MMP-9. We added this point to the text (line 103-105).

  • Results: I suggest the authors to better organize the tables/figures reporting only the significant findings.

Response: Thank you for raising this point, according to which we organized the Tables/Figures to report only significant results. We deleted the subcategory analysis of patients with exanthem subitum from Figure 1D-1F and 4D-4F. We also eliminated Figure 4A-4C.

  • Discussion: the authors should better discuss their results in a more organized manner.

Response: We appreciate this advice, according to which we reorganized the Discussion section. Specifically, we deleted the subcategory analysis of patients with exanthem subitum from the Discussion section.

  • Please correct typos and English language throughout the manuscript (mainly in the discussion).

Response: Thank you for your advice. Prior to the first submission, the manuscript underwent English language editing. The revised manuscript has been re-checked by a reliable proofreading company.

Round 2

Reviewer 1 Report

The resubmission of this manuscript is improved in comparison to the initial submission.  While some of my comments could not be directly addressed due to sample limitations, most of my concerns were addressed in some manner.  I have one remaining significant concern with this submission:

1) The discussion for this manuscript is brief and only introduces a few new references.  The discussion should be lengthened more to fully discuss the results of the data and its relevance to the field and other neurological diseases.  For example, the authors briefly describe the role of MMP9 and the blood-brain barrier.  This should be expanded and described in greater detail in the context of AESD.  In addition, none of the cytokine data is discussed either.  This needs to be added to improve the quality of this manuscript.

Author Response

Responses to the comments of Reviewer 1

We appreciate your critical and constructive comments, which have helped us to improve the quality of the revised manuscript.

  • The discussion for this manuscript is brief and only introduces a few new references. The discussion should be lengthened more to fully discuss the results of the data and its relevance to the field and other neurological diseases. For example, the authors briefly describe the role of MMP9 and the blood-brain barrier. This should be expanded and described in greater detail in the context of AESD. In addition, none of the cytokine data is discussed either. This needs to be added to improve the quality of this manuscript.

Response: Thank you for raising this point. We agree that the discussion for this manuscript is brief. The main reason for this point is that only a few studies about cytokines, MMP-9, and TIMP-1 in patients with AESD had been published. Notably, there are no reports about the relationship between MMP-9 and blood-brain barrier in patients with AESD other than the cited study in our manuscript (reference number, [22]). Therefore, we cannot add the further discussion or previous reports about the relationship between MMP-9 and blood-brain barrier in patients with AESD, which was recommended by your advice. Furthermore, there are no reports about the MMP-9-independent increase of TIMP-1 as described in the manuscript. However, we think that the rarity of previous reports about MMP-9 or TIMP-1 in patients with AESD is the strong point of our study, because the present study can provide novel insights into this field that will prompt future studies using large number of patients.

On the other hand, there are a few studies, which had only small sample size, or case reports about cytokine and chemokine profiles in patients with AESD. As indicated, we added the following sentences to the manuscript (line 296-301): “In children with AESD due to exanthem subitum, Kawamura et al. analyzed cytokine and chemokine profiles in serum and CSF: significant increases in serum concentrations of IL-10 and significant decrease in CSF concentrations of IL-1β were observed. Additionally, a few studies or case reports showed the increases of serum concentrations of IL-6 and -10 and tumor necrosis factor-α in patients with AESD, although the cause of increases of these cytokines had not been demonstrated” (lines 296-302).

Furthermore, as indicated, we added the following references to discussion:

  1. Kawamura Y, Yamazaki Y, Ohashi M, Ihira M, Yoshikawa T. Cytokine and chemokine responses in the blood and cerebrospinal fluid of patients with human herpesvirus 6B-associated acute encephalopathy with biphasic seizures and late reduced diffusion. J Med Virol 2014, 86, 512-518.
  2. Yamaguchi H, Nagase H, Ito Y, Matsunoshita N, Mizutani M, Matsushige T, Ishida Y, Toyoshima D, Kasai M, Kurosawa H, et al. Acute focal bacterial nephritis characterized by acute encephalopathy with biphasic seizures and late reduced diffusion. J Infect Chemother 2018, 24, 932-935.
  3. Shiba T, Hamahata K, Yoshida A. Acute encephalopathy with biphasic seizures and late reduced diffusion in Kawasaki disease. Pediatr Int 2017, 59, 1276-1278.
  4. Ichiyama T, Suenaga N, Kajimoto M, Tohyama J, Isumi H, Kubota M, Mori M, Furukawa S. Serum and CSF levels of cytokines in acute encephalopathy following prolonged febrile seizures. Brain Dev 2008, 30, 47-52.

Reviewer 2 Report

The authors did their best to address my concerns. I think that the manuscript now can be suitable for publication.

Author Response

Responses to the comments of Reviewer 2

We appreciate your help with reviewing our manuscript.
